# Cancer Incidence and Mortality among Petroleum Industry Workers and Residents Living in Oil Producing Communities: A Systematic Review and Meta-Analysis

**DOI:** 10.3390/ijerph18084343

**Published:** 2021-04-20

**Authors:** Felix M. Onyije, Bayan Hosseini, Kayo Togawa, Joachim Schüz, Ann Olsson

**Affiliations:** Environment and Lifestyle Epidemiology Branch, International Agency for Research on Cancer (IARC/WHO), 150 Cours Albert Thomas, CEDEX 08, 69372 Lyon, France; hosseinib@students.iarc.fr (B.H.); TogawaK@iarc.fr (K.T.); SchuzJ@iarc.fr (J.S.); olssona@iarc.fr (A.O.)

**Keywords:** systematic review, meta-analysis, petroleum industry, occupational exposure, environmental pollution, neoplasms

## Abstract

Petroleum extraction and refining are major sources of various occupational exposures and of air pollution and may therefore contribute to the global cancer burden. This systematic review and meta-analysis is aimed at evaluating the cancer risk in petroleum-exposed workers and in residents living near petroleum facilities. Relevant studies were identified and retrieved through PubMed and Web of Science databases. Summary effect size (ES) and 95% confidence intervals (CI) were analysed using random effect models, and heterogeneity across studies was assessed (I^2^). Overall, petroleum industry work was associated with an increased risk of mesothelioma (ES = 2.09, CI: 1.58–2.76), skin melanoma (ES = 1.34, CI: 1.06–1.70 multiple myeloma (ES =1.81, CI: 1.28–2.55), and cancers of the prostate (ES = 1.13, Cl: 1.05–1.22) and urinary bladder (ES = 1.25, CI: 1.09–1.43) and a decreased risk of cancers of the esophagus, stomach, colon, rectum, and pancreas. Offshore petroleum work was associated with an increased risk of lung cancer (ES = 1.20; 95% CI: 1.03–1.39) and leukemia (ES = 1.47; 95% CI: 1.12–1.92) in stratified analysis. Residential proximity to petroleum facilities was associated with childhood leukemia (ES = 1.90, CI: 1.34–2.70). Very few studies examined specific exposures among petroleum industry workers or residents living in oil producing communities. The present review warrants further studies on specific exposure levels and pathways among petroleum-exposed workers and residents living near petroleum facilities.

## 1. Introduction

The petroleum industry has been, and still is, an important pillar in many countries′ economy [1]. Petroleum (crude oil) is the origin of many complex mixtures such as petrol and diesel fuel and is, via chemical processes, used to produce plastics and other useful materials such as textiles [2], pesticides, cosmetics, paints, and insulating materials [3]. Workers in the petroleum industry are exposed to a variety of known or potentially harmful substances [4] and emissions from oil and gas extractions are among the major sources of air pollution in environments and communities where such facilities are situated and operate [5,6,7]. These potential harmful exposures include, among others, heavy metals and hydrocarbons such as benzene which is up to 4 g/L in crude oil [8]. Asbestos which is known to be the cause for mesothelioma and cancers of the lung, larynx, and ovary has been used extensively in petroleum industries [9]. Both benzene and asbestos are classified as Group 1 carcinogens by the International Agency for Research on Cancer (IARC) [4].

A recent systematic review and meta-analysis including studies published up to 2012 was based on data from 36 cohort studies assessed the meta-relative risk of 11 cancers among petroleum refinery workers [10]. Their results showed elevated meta-relative risks of mesothelioma, malignant skin melanoma, and acute lymphoid leukemia and no elevated risks of total leukemia, acute non-lymphocytic leukemia, chronic lymphocytic leukemia, chronic myeloid leukemia, multiple myeloma, non-Hodgkin’s lymphoma (NHL), and lung and kidney cancers. The most recent review of residents near petroleum industry sites has also found elevated risks for leukemia (the review only focused on hematological malignancy) [11].

The aim of this present systematic review was to evaluate the epidemiological studies on cancer risks among petroleum workers and residents living near petroleum facilities through articles published between 1990 and 2019. Our review adds to the previous review [10] with four cohort, 14 case-control and two cross sectional studies on petroleum workers and with 13 additional cancer sites (i.e., cancers of the brain, urinary bladder, esophagus, stomach, colon, rectum, liver, pancreas, gall bladder, prostate, testis, breast and Hodgkin’s lymphoma).

## 2. Materials and Methods

### 2.1. Eligibility Criteria

Our systematic review was conducted in accordance with the Preferred Reporting Items for Systemic Reviews and Meta-Analysis (PRISMA) check list of 2009 [12]. The present study included original articles in English published in peer reviewed journals from January 1990 to December 2019 (30 years period). The vast majority of relevant literature was published within these three decades, including early studies which were updated during that time. Very few studies were published only in the 1980s with working circumstances rather diluting the overall picture. The studies had to be cohort, case-control, or cross-sectional studies on cancer in petroleum workers or residents living close to operating facilities. Only studies that provided estimates of the Relative Risk (RR), such as Standard Mortality Ratios (SMR), Standard Incidence Ratios (SIR), Incidence Rate Ratio (IRR), Mortality Rate Ratio (MRR) or Odds Ratio (OR) with 95% confidence intervals (CI) were included. When multiple studies were identified from the same cohort/authors, we included them if they reported results for separate cancer sites or for different geographical locations, but if they studied the same cancer site in the same geographical location, the latest result with the longest follow-up or the largest study population was included.

### 2.2. Information Sources

Scientific papers were identified and retrieved through PubMed and Web of Science (WOS) databases, imported and automatically screened for duplicates in EndNote version X8.2, and later screened manually. The search strategies included a list of key words and Mesh terms (Appendix A). Additional relevant articles were identified through the exploration of lists of references. The initial search was performed in December 2019. Thereafter search alerts were placed on both databases using the search strategy to enable regular updates, with relevant articles until February 2021. We defined petroleum facilities as the upstream sector including exploration, development, and production of petroleum resources; the downstream sector including refining, marketing, and distribution of petroleum.

### 2.3. Assessment of Eligibility, Quality and Data Extraction

Eligibility screening of title and abstract were carried out independently by the first and second authors, and disagreements were resolved by the last author through discussions in line with the Cochrane handbook for systematic reviews [13]. Following removal of duplicates, the following data were extracted from the full-text articles: author and year of publication, country, continent, minimum employment time (petroleum industry workers), follow-up duration, exposed population, specific exposure, cancer site/type, number of cases and controls, as well as RR with their respective CIs. Information on study design (cohort, case-control and cross-sectional) was also extracted. All papers selected for inclusion were subjected to a rigorous appraisal for methodological quality using Joanna Briggs Institute critical appraisal (JBI) tools for cohort, case-control and cross-sectional studies [14]. The critical appraisal checklist has 11 criteria for cohort, 10 for case-control and 8 for cross-sectional studies, each question with “yes” score 1, “no” score 0 and “unclear” score 0.

### 2.4. Statistical Analyses

Random effects meta-analyses were conducted using the Stata^®^, version 14.0. (Stata^®^ statistical software, College Station, TX, USA) “metan” command in order to (1) compute pooled effect size (ES) for mortality studies and incidence studies with 95% confidence intervals (CI) and (2) explore heterogeneity between studies, expressed as a percentage (I^2^) [15]. Heterogeneity was considered as “no heterogeneity” when I^2^ was 0%, “probably unimportant” when I^2^ was 1% to 35%, “moderate” when I^2^ was 36% to 55%, “substantial” when I^2^ was 56% to 70% and “considerable” when I^2^ was 71% to 100% [16]. Mortality and incidence studies were combined to one pooled ES for some digestive and hematological cancers (Figure 2) because of similarity in incidence and mortality rates of the underlying cancer types and due to few studies. Likewise, the relative risks from incidence and mortality studies among residents were combined except for brain and skin melanoma. Potential publication bias was evaluated using the Egger’s funnel plot and its associated *p*-value [17,18]. We also performed stratified analysis for the different categories of petroleum industries i.e., refinery, petroleum, petrochemical, refinery and petrochemical, and offshore. In all instances for this stratified analysis both mortality and incidence were combined.

Meta regression (metareg in Stata^®^) was used to determine the impact of specific variables (country, continent, decade (1990–1999/2000–2009/2010–2019)) on the study ES. For heterogeneity and Egger’s regression tests a *p*-value less than 0.05 were considered statistically significant.

## 3. Results

### 3.1. Search Strategy Outcome

The search strategy for PubMed, WOS and other records yielded a total of 969 studies, whereof 273 papers were duplicates (see the PRISMA flow chart in Figure 1). Following duplicate removal, 569 non-eligible papers were removed based on titles and abstracts. One hundred and twenty-seven papers underwent full-text assessment for eligibility, and resulted in 57 eligible papers (41 cohort, 14 case-control and two cross-sectional studies) from which the data were extracted. Of those, 41 cohort and 10 case-control studies were included in the meta-analysis, while four case-control and two cross-sectional studies were not included in the meta-analysis because these studies examined specific cancer types or exposures for which there were only one study and therefore it was not possible to perform a meta-analysis; yet they are described and discussed separately within our review (Table 1 and Table 2).

### 3.2. Study Characteristics

Table 1 and Table 2 shows the list of 57 eligible papers with the majority of studies coming from North America (*n* = 23) and without a single study from Africa. In terms of publication year, most articles (*n* = 31) were published in the 2000s. Out of the 57 studies, 43 estimated cancer risk among workers, while 14 estimated cancer risk among residents.

### 3.3. Assessment of Risk of Bias

The methodological quality of studies evaluated on petroleum workers resulted in an average score of 74% (Cohort studies) and 84% (Case control studies). The average score of papers on residents living near petroleum facilities was 41% (Cohort studies), 61% (Cross-sectional studies) and 70% (case-control studies). We did not exclude papers on residents based on their methodological quality due to very few papers being eligible for the final analysis in this group. The risk of bias or quality assessment grading for the different components of each study is shown in Appendix A.

### 3.4. Review and Quantitative Analyses of Petroleum Workers

In this section we describe 36 papers with results from cohort studies among petroleum workers (summarized in Figure 2).

#### 3.4.1. Respiratory Cancers

##### Lung Cancer

Combining 18 mortality studies [21,25,27,28,29,31,35,36,37,38,39,40,43,44,45,49,50,52] we observed a pooled ES of 0.82 (95% CI: 0.76, 0.88) based on 7231 lung cancer deaths with considerable heterogeneity across studies. When excluding the only study with a significantly elevated SMR of 1.69 (95% CI: 1.03, 2.61), i.e., “Stenehjem”, the heterogeneity remained “considerable” (84.8%). Similarly, the seven incidence studies [19,20,24,32,33,42,48] showed a pooled ES of 0.81 (95% CI: 0.16,1.06) based on 527 incident cases, again with considerable heterogeneity (I^2^ = 78, *p* ≤ 0.01) (Figure 2). The meta-regression analysis did not reveal any determinants of the heterogeneity, e.g., decade and continent were not significantly associated with ES (*p*-values = 0.99 and 0.54, respectively). The funnel plot did not demonstrate significant asymmetry and Egger’s regression test did not show evidence of publication bias (Egger, *p* = 0.99) (Appendix A).

##### Mesothelioma

There were nine mortality studies [20,25,26,28,29,37,39,40,50] of mesothelioma which yielded a pooled ES of 1.98 (95% CI: 1.46, 2.67) based on 280 mesothelioma deaths. All studies but one on distribution workers Sorahan [37] showed elevated mortality. There was considerable heterogeneity (I^2^ = 72, *p* ≤ 0.01). However, the heterogeneity decreased I^2^ = 55.7% and the ES increased to 2.24 (95% CI 1.75, 2.88) when the “distribution workers” in Sorahan’s study was dropped. In the same vein, four incidence studies [9,20,32,33] which included a total of 52 incident cases showed a pooled ES of 2.09 (95% CI: 1.58, 2.76). All four incidence studies showed elevated risk, and there was substantial heterogeneity (I^2^ = 61, *p* = 0.69) in ES which was not associated with decade (*p*-value = 0.63) or country (*p*-value = 0.18). Combining the incidence and mortality studies resulted in an asymmetric funnel plot but did not show evidence of publication bias (Egger, *p* = 0.99) see Appendix A.

#### 3.4.2. Skin Cancers

##### Skin Melanoma

The pooled ES from 13 mortality studies [21,25,27,28,37,38,40,43,44,46,49,50,52] was 1.11 (95% CI: 0.93,1.32) based on 319 skin melanoma deaths with “moderate” heterogeneity. Half of the studies had an ES below 1.00. However, the 7 incidence studies [19,20,21,32,33,42,48] which included a total of 137 cases yielded a pooled ES of 1.34 (95% CI: 1.06,1.70), but with substantial heterogeneity (I^2^ = 59, *p* = 0.02). The symmetric funnel plots and Egger’s test suggest no publication bias (*p* = 0.66 for mortality and 0.83 for incidence studies).

##### Skin Cancer (Non-Specific)

A meta-analysis of eight studies [20,21,29,30,35,37,50,51] of skin cancer mortality showed no elevated risk (pooled ES = 0.95 (95% CI: 0.81, 1.10)) based on 137 skin cancer deaths, with no heterogeneity (Figure 2). However, one cohort study [30], which was conducted in Korea, estimated a three-fold risk. We found no evidence of publication bias (Appendix A).

#### 3.4.3. Urinary Tract Cancers

##### Urinary Bladder Cancer

Nine mortality studies [20,21,25,27,29,30,37,50,51] including 435 urinary bladder cancer deaths resulted in a pooled ES of 0.91 (95% CI: 0.79, 1.04), with no heterogeneity between studies. In contrast, the pooled ES for incidence studies (*n* = 6) [19,20,21,33,42,48] was 1.25 (95% CI: 1.09, 1.43), based on 190 incident urinary bladder cancers. There was no heterogeneity between studies, and the funnel plot and Egger’s test indicated no potential publication bias (Egger’s test *p* = 0.50 and 0.34 for mortality and incidence studies) (Appendix A).

##### Kidney Cancer

Combining 14 mortality studies [20,25,27,36,37,38,40,43,44,45,49,50,51,52] yielded a pooled ES of 1.03 (95% CI: 0.91, 1.16) based on 537 kidney cancer deaths, and 1.04 (CI: 0.81–1.32) for six incidence studies [19,20,21,23,42,48] based on 60 cancer cases. There was no heterogeneity for the incidence studies, and “probably unimportant” heterogeneity for the mortality studies (Figure 2 and Appendix A). Egger’s regression test did not show evidence of publication bias in either mortality or incidence studies (*p* = 0.18 and 0.08).

#### 3.4.4. Reproductive System Cancers

##### Prostate Cancer

Effect size of 0.97 (95% CI: 0.91, 1.04) was observed for prostate cancer mortality based on 13 mortality studies [20,21,25,29,30,35,37,40,43,45,49,50,51] including 1408 prostate cancer deaths, with “probably unimportant” heterogeneity. The pooled ES for incidence studies (*n* = 7) [20,21,23,32,42,48] was 1.13 (95% CI: 1.05, 1.22) based on 345 incident prostate cancer, with no substantial heterogeneity (Appendix A). The only incidence study with ES below 1.00 was the study of Lewis et al. (0.6; 95% CI: 0.41, 1.03) (Lewis et al., 2003).

##### Testicular Cancer

The pooled ES was 0.66 (95% CI: 0.32, 1.35) based on a study [21] from Australia (1.01; 95% CI: 0.03, 5.63, 1 case) and a study [37] from the United Kingdom (UK) (0.40; 95% CI: 0.11, 1.03, four cases for refinery workers and 0.92; 95% CI: 0.30, 2.15, four cases for distribution workers). The pooled ES for four incidence studies [20,21,23,42] was 1.07 (95% CI: 0.74, 1.54) based on 20 incident testicular cancers, with no heterogeneity across studies.

##### Breast Cancer

In five mortality studies [25,37,38,43,50] we observed a pooled ES of 0.98 (95% CI: 0.77, 1.25) based on 77 breast cancer deaths (male 14 and female 63), with “probably unimportant” heterogeneity. Egger’s regression test did not show evidence of publication bias (Egger, *p* = 0.75) (Appendix A).

#### 3.4.5. Digestive and Accessory Digestive organ Cancers

##### Esophageal and Gastric Cancers

Esophageal cancer was examined in nine mortality studies [20,21,25,31,37,43,44,49,50], of which only one study ([44], 1.16; 95% CI: 0.82, 1.61) reported a slightly increased but non-significant risk. The pooled ES showed a reduced relative risk of 0.84 (95% CI: 0.74, 0.95) based on 493 esophageal cancer deaths, with “probably unimportant heterogeneity”. Similarly, the pooled ES for stomach cancer was less than 1.0 (0.88; 95% CI: 0.79, 0.98) based on twelve studies (mortality = 11 and incidence = 1) [20,21,25,29,31,37,43,44,48,49,50,51] including a total of 983 gastric cancer deaths and one incident case. There was a “moderate” heterogeneity with no significant *p*-value (Figure 2).

##### Colon and Rectum Cancers

Colon cancer was reported in 14 studies (mortality *n* = 12 and incidence *n* = 2) [20,21,25,29,31,37,42,43,44,48,49,50,51,52] and the pooled ES was 0.85 (95% CI: 0.80–0.90) based on 1325 colon cancer deaths and 45 incident cases with no heterogeneity across studies. There were 10 mortality studies that reported on rectum cancer among petroleum workers, whereof only one study reported non-significantly elevated risk 1.16 but with a large confidence interval (95% CI: 0.32, 2.96). The pooled ES of cancer mortality studies (*n* = 10) [20,25,29,37,43,44,45,49,50,52] of the rectum was 0.89 (95% CI: 0.80, 0.98) based on 459 rectal cancer deaths, with no heterogeneity (Figure 2). Egger’s regression test (Egger, *p* = 0.30) with funnel plot did not show evidence of publication bias for estimates of digestive cancer studies (Appendix A).

##### Liver, Pancreas and Gall Bladder Cancers

Mortality of liver cancer was reported in 10 studies [20,21,25,29,31,37,44,45,49,50], with only one study [37] showing an estimated risk above 1.00. i.e., 1.20 (95% CI: 0.80, 1.72). The pooled ES was 0.73 (95% CI: 0.63, 0.86) based on 265 liver cancer deaths, with “probably unimportant” heterogeneity I^2^ = 22.2%, *p*-value = 0.23. Pancreatic cancer mortality was reported in 12 studies [20,21,25,29,31,37,43,44,48,49,50,51] of which three studies had an ES above 1.00. The pooled ES was 0.88 (95% CI: 0.82, 0.96) based on 698 pancreatic cancer deaths, with “probably unimportant” heterogeneity. Based on the four gall bladder cancer mortality studies [20,21,23,37] including 58 gall bladder cancer deaths, the pooled ES was 1.38 (95% CI: 0.78, 2.45), with a “moderate” heterogeneity (Figure 2). Three of the studies reported a risk estimate above 1.00, of which one (Lewis et al. 4.28; 95% CI: 1.17, 10.95) was significant. The symmetric funnel plot for accessory digestive organs did not suggest publication bias, which was also supported by Egger’s regression test (Egger, *p* = 0.98) (Appendix A).

#### 3.4.6. Central Nervous System

##### Brain Cancer

The pooled ES from 17 mortality studies [20,21,22,27,31,35,36,37,38,42,46,47,48,49,50,51,52] was 1.04 (95% CI: 0.95, 1.12) based on 662 brain cancer deaths, with no heterogeneity between studies. The estimates (two SMR and one SIR) in three studies conducted in the USA and Italy [27,31,42] were elevated, while the pooled ES from three incidence studies [19,31] was 0.73 (95% CI: 0.28, 1.92) based on 10 brain cancer cases with “probably unimportant” heterogeneity. Combination of mortality and incidence studies did not show publication bias with Egger’s regression test (Egger, *p* = 0.63) (Appendix A).

#### 3.4.7. Hematological Cancers

##### Hodgkin’s Lymphoma

Thirteen studies [25,27,28,29,36,37,38,43,44,49,50,51,52] reported risk estimates for mortality for Hodgkin’s lymphoma (HL) with six of the studies having estimated risk of 1.00 and above. The pooled ES was 1.03 (CI: 0.86–1.25) based on 123 HL deaths, with no heterogeneity.

##### Non-Hodgkin’s Lymphoma

The pooled ES for the 10 mortality studies [20,21,25,27,28,29,44,50] was 0.95 (95% CI: 0.85, 1.05) based on 369 NHL deaths. There was no heterogeneity for mortality studies. The pooled ES from seven incidence studies [19,20,23,31,41,42,48] was 1.17 (95% CI: 0.87, 1.58) based on 79 incident NHLs with “moderate” heterogeneity (Figure 2). The majority of the mortality studies showed a reduced risk, while the opposite was observed for the incidence studies, although both pooled analyses resulted in wide confidence intervals. The funnel plot for mortality and incidence studies combined did not demonstrate significant asymmetry and Egger’s regression test did not show evidence of publication bias (Egger, *p* = 0.53) (Appendix A).

##### Multiple Myeloma

Nine mortality studies [20,21,25,28,29,36,37,44,50] resulted in a pooled ES of 1.04 (95% CI: 0.89, 1.21) based on 264 multiple myeloma (MM) deaths. The heterogeneity was “probably unimportant” (Figure 2). In contrast, five incidence studies [19,20,34,48,50] had a pooled ES of 1.80 (95% CI: 1.28, 2.55) based on 32 incident MMs, with no heterogeneity. The five incidence studies showing an increased risk were conducted in the USA and Australia, with the highest risk estimate observed in the smallest study of less than 1500 workers and four cases [19]. There was no publication bias using Egger’s regression for mortality and incidence studies combined (Egger, *p*= 0.16).

##### Leukaemia and Its Subtypes

A combined (mortality and incidence) nine acute myeloid leukemia (AML) studies [19,20,21,25,32,34,37,44,50] had a pooled estimate of 1.11 (95% CI: 0.91, 1.36) based on 217 AML deaths and 23 incident AMLs, with “moderate” heterogeneity. Five combined studies [20,21,34,44,50] of chronic myeloid leukemia (CML) also had a pooled ES of 1.06 (95% CI: 0.91, 1.36) based on 38 CML deaths and three incident CMLs, with “no heterogeneity”. The same 5 studies also reported acute lymphocytic leukemia (ALL) with a pooled ES of 1.06 (95% CI: 0.63, 1.78) based on 17 ALL deaths and 1 incident ALL, with no heterogeneity. Four combined studies [25,34,44,50] on chronic lymphocytic leukemia (CLL) yielded a pooled estimate of 0.97 (95% CI: 0.73, 1.29) based on 49 CLL deaths and one incident CLL, with “no heterogeneity”. However, there were 22 studies [19,21,23,25,27,28,31,32,35,36,37,38,40,41,43,44,45,49,50,51,52] that reported total leukemia with a pooled ES of 1.08 (95% CI: 0.97, 1.19) in a combined (mortality and incidence) study based on 684 leukemia deaths and 23 incident cases with “moderate” and statistically significant heterogeneity (I^2^ = 36.7 %, *p* = 0.03 (Figure 2). The leukemia-subtypes were not part of the overall leukemia results and out of the 22 studies, 13 had ES > 1.00. The funnel plot and Egger’s regression test indicated there was no publication bias for leukemia (*p* = 0.59) and its subtypes (*p* = 0.59).

#### 3.4.8. Stratified/Subgroup Analysis

Among the five categories of petroleum industries evaluated, there was no substantial effect on the summary ES results of the main meta-analysis (Table 3). Exceptions were the significant elevation of lung cancer (OR 1.20; 95% CI: 1.03, 1.39) and leukemia (combined) (OR 1.47; 95% CI: 1.12,1.92) among offshore workers. Mesothelioma was significantly elevated among four categories evaluated i.e., refinery (OR 1.94; 95% CI: 1.00,3.76), petroleum (OR 1.58; 95% CI: 1.30–1.93), refinery and petrochemical (OR 2.86; 95% CI: 2.16, 3.77), and offshore (OR 2.47; 95% CI: 1.66, 3.67). Skin melanoma was significantly elevated in the petroleum category (OR 1.28; 95% CI: 1.10,1.50).

### 3.5. Review and Quantitative Analyses of Residents Living Near Petroleum Facilities

Herein, we describe five papers including results from cohort studies among residents living near petroleum plants by cancer site (summarized in Figure 3).

#### 3.5.1. Solid Cancers

Lung cancer was reported in three studies [61,63,66], out of which one study conducted in Ecuador (Kelsh et al.) showed a reduced risk of mortality among residents living near petroleum facilities (MRR = 0.49; 95% CI: 0.39, 0.63). The pooled ES was 0.88 (95% CI: 0.64, 1.21), based on 830 lung cancer deaths and 77 incident cases, and there was considerable heterogeneity (I^2^ = 89, *p* ≤ 0.01).

Skin melanoma incidence risks in two studies [61,63] had a pooled ES of 0.93 (95% CI: 0.01, 94.98) based on nine incident cases and three deaths. The studies showed individual relative risks at opposite extremes; a study by Hurtig and Sebastian [61] showed a RR of 10.15 (95% CI: 2.19, 46.97), while Kelsh et al. found an inverse association (MRR 0.09; 95% CI: 0.03, 0.27) [63].

Brain cancer was investigated in two studies [61,63] from Ecuador which compared brain cancer risks in residents living near petroleum facilities with the national rate. Hurtig and Sebastian, reported RR 3.80 (95% CI: 0.24, 60.65) for females and RR 0.14 (95% CI: 0.01, 1.34) for males [61]. The study by Kelsh et al. reported an MRR of 0.31 (95% CI: 0.21, 0.47). The pooled ES was 0.39 (95% CI: 0.11, 1.43) based on 24 brain cancer deaths and two brain cancer cases, with a “moderate” heterogeneity across studies (Figure 3 and Appendix A).

#### 3.5.2. Hematological Cancers

Multiple myeloma was reported in two studies [63,66] showing a pooled estimate of 0.85 (95% CI: 0.29, 2.50) with considerable heterogeneity (I^2^ = 91, *p* ≤ 0.01). The study conducted in Ecuador showed a decreased risk of multiple myeloma mortality (MRR 0.12; 95% CI: 0.04, 0.36, 3 cases) whereas the study conducted in Baglan Bay, UK, reported excess mortality risks in relation to distance to petroleum facilities as SMR 2.15 (95% CI: 1.25, 3.67, 13 cases) for residents living 3.0 km away and SMR 1.49 (95% CI: 1.10, 2.01, 42 cases) for 7.5 km [66]

Hodgkin’s lymphoma (HL) mortality was examined in one study [66] The authors reported estimates of 1.27 (95% CI: 0.26, 3.71,3 cases) for 3.0 km radius away from petroleum facilities and 1.09 (95% CI: 0.64, 1.86, 11 cases) for 7.5 km radius away from petroleum facilities in Baglan Bay in the UK.

Non-Hodgkin’s lymphoma was evaluated in three studies [63,64,66] conducted in Spain (RR 1.09; 95% CI: 0.97, 1.24, 614 cases for males and RR 1.12 CI: 0.99–1.27, 675 cases for females) [64], Ecuador (MRR 0.40, 95% CI: 0.27, 0.58, 27 cases) [63] and in the UK (SMR 1.09; 95% CI: 0.97, 1.24, 12 cases for 3.0 km radius away from petroleum facilities and SMR 1.07; 95% CI: 0.61, 1.87, 51 cases for 7.0 km radius) [66]. The pooled ES was 0.91 (95% CI: 0.71, 1.17). There was considerable heterogeneity (I^2^ = 85, *p* ≤ 0.01) (Figure 3).

Leukaemia risk was evaluated in three studies [63,65,66], Hurtig and Sebastian reported the highest risk associated with petroleum exposure (RR 2.56; 95% CI: 1.35, 4.86, 28 cases) in a study conducted in Ecuador. Similarly, in the same country the lowest risk estimate [63] was reported (MRR 0.53; 95% CI: 0.43, 0.58, 99 cases). The other studies were in Sweden (SIR 0.89; 95% CI: 0.66, 1.18, 50 cases) [65] and in the UK (SMR 0.85; 95% CI: 0.41, 1.57, 10 cases for 0.3 km radius away from petroleum facilities and SMR 1.14; 95% CI: 0.88, 1.45, 61 cases for 0.7km radius) [66]. The pooled ES was 0.99 (95% CI: 0.65, 1.57). There was a considerable heterogeneity (I^2^ = 91, *p* ≤ 0.01). On the basis of funnel plots and Egger’s tests, we observed no indication of publication bias (*p* = 0.76) (Figure 3 and Appendix A).

### 3.6. Review and Quantitative Analyses of Case-Control Studies of Workers and Residents Living Near Petroleum Facilities

Hereunder we describe 10 case-control studies among workers and residents living near petroleum facilities by cancer site (summarized in Figure 4).

#### 3.6.1. Petroleum Workers

##### Hematopoietic Cancers

Leukaemia overall was examined in two case-control studies [54,59] and yielded a pooled ES of 1.54 (95% CI: 0.23, 10.44, four cases and 59 controls) for cumulative benzene exposure and 0.90 (95% CI: 0.49, 1.64, 23 cases and 135 controls) for maximum benzene intensity, with no heterogeneity.

In two case-control studies [56,58] on leukemia subtypes, we observed an elevated risk of AML with a pooled ES of 2.02 (95% CI: 0.66, 6.19) based on 70 cases and 656 controls in cumulative benzene exposure analysis on highest compared to lowest. There was a “moderate” but non-significant heterogeneity (I^2^ = 42.6%, *p* = 0.19). The pooled ES in relation to maximum intensity was 2.17 (95% CI: 0.86, 5.48) with “probably unimportant” heterogeneity (I^2^ = 23.3%, *p* = 0.25 (Figure 4). Both studies (Rushton et al. = 1.39; 95% CI: 0.68, 2.85 and Stenehjem et al. = 4.85; 95% CI: 0.88, 27.00) had estimates above 1.00 with a wide CI. For CLL, the pooled ES was 1.77 (95% CI: 0.40, 7.89) in relation to cumulative benzene exposure, with a “moderate” heterogeneity (I^2^ = 55.9%, *p* = 0.13) (Figure 4 and Appendix A).

For chronic myeloid leukemia (CML) Glass et al. [57] reported single increased ORs (cumulative benzene exposure = 2.20; 95% CI: 0.63, 7.68 and maximum benzene exposure intensity = 2.12; 95% CI: 0.77, 5.82) out of many cases in a multi-country nested case-control study including 28 cases and 122 matched controls, but the authors concluded that no convincing association was identified between CML and low exposure to benzene because there were no coherent patterns.

##### Solid Cancers

Kidney cancer in relation to cumulative benzene exposure was analysed in two nested case-control studies [55,60] with a pooled ES of 1.55 (95% CI: 0.84, 2.83, based on 130 cases and 487 controls) with no heterogeneity. This is similar to the pooled ES of the cohort studies where both mortality and incidence have OR above 1.00

Finkelstein [53] conducted a death certificate-based case-control study of 17 mesothelioma cases and 424 lung cancer cases compared to other blue-collar workers in order to investigate risks among refinery and petrochemical sector workers in Canada. Employment as a maintenance worker in the refinery and petrochemical company was associated with an OR of 24.5 (90% CI: 3.1, 102) for mesothelioma, and 1.73, 90% CI: 0.83, 3.6) for lung cancer.

#### 3.6.2. Residents Living Near Petroleum Facilities

##### Childhood Leukaemia and Other Hematopoietic Cancers

The two studies [70,74] yielded a significantly increased pooled ES of 1.90 (95% CI: 1.34, 2.70) with no heterogeneity between studies, *p* = 1.00. Weng et al. conducted a case-control study including 405 childhood leukemia deaths showing an elevated risk of 1.90 (95% CI: 1.26, 2.87) for children living in the municipalities characterized by the highest proportion of the population employed in the petrochemical industry, when compared to the municipality with the lowest proportion. McKenzie et al. [74] conducted a registry-based case control study in rural Colorado in the USA, with study participants 0–24 years old and diagnosed with cancer between 2001 and 2013; of which were 87 children/young adults with ALL and 50 with NHL. The results showed that ALL cases 5–24 years old were more likely to live in the highest tertile of residential proximity to oil and gas development, with a significant test for trend (*p*-value = 0.035). No association was found for younger ALL cases or NHL cases.

Lyons et al. [68] compared the incidence of leukemia and lymphoma in young people (0–24 years) living in the surroundings (within 1.5 and 3 km) of a petrochemical plant with the expected numbers of cases calculated from the Welsh cancer registration rates. The authors concluded that the incidence of leukemia and lymphomas in young people was not significantly greater than expected.

Micheli et al. [73] investigated mortality from hematological malignancies (HM) among residents in relation to benzene pollution in three municipalities surrounding an Italian refinery, based on 177 HM-related deaths. The authors observed an excess risk of death in relation to proximity of residence to refinery for women and in a sub-group of study participants that were retired, homemakers, or unemployed, but not in men. The authors suggested that the positive association for females in this subgroup was due to them supposedly spending most of their time at home and having, therefore, the highest exposure due to residential proximity to the refinery [73].

##### Solid Cancers

Yu et al. [72] assessed the association between residential petrochemical exposure and brain tumor risk in a population-based case-control study including 143 brain tumor cases residing in an area of ∽657.1 km^2^ hosting four petrochemical industrial complexes in Kaohsiung, China/Taiwan [72]. The results showed no association between residential exposure to petrochemicals and brain tumor risks (OR 1.00; 95% CI: 0.83, 1.12). On the contrary, a case-control study by [71], which included 340 brain cancer cases, found that those who lived in the municipalities characterized by the highest levels of petrochemical air pollution had a statistically significant higher risk of developing brain cancer (OR 1.97; 95% CI: 1.35, 2.88). The meta-analysis of both studies resulted in a pooled ES of 1.37 (95% CI: 0.71, 2.66) with substantial and significant heterogeneity (I^2^ = 90.6%, *p* ≤ 0.01)

Tsai et al. conducted a death certificate-based case-control study on bladder cancer in China/Taiwan from 1995 through 2005, including 821 cases. The results showed that subjects who lived in petrochemical polluted environment had an increased risk of bladder cancer with an OR of 1.65 (95% CI: 1.22, 2.31).

Choi et al. [67] conducted a cross-sectional study of 63042 participants to evaluate prostate cancer risk in relation to oil spillage in Taean County. They reported elevated incidence rate ratio (IRR = 1.5; 95% CI: 0.8, 2.7) for the high exposed area compared to the low exposed area (IRR= 0.8; 95% CI: 0.5, 1.4).

## 4. Discussion

Our systematic review and meta-analyses included 43 publications on cancer risks among petroleum workers and 14 among residents living near petroleum industry sites. Among petroleum workers, six cancer types were associated with elevated risks, namely mesothelioma in both incidence and mortality studies, as well as multiple myeloma, skin melanoma, prostate cancer and urinary bladder cancer in incidence (but not mortality) studies. In stratified analysis, lung cancer and leukemia were associated with an elevated risk among offshore workers. Cancers of the liver, esophagus, colon, pancreas, stomach, and rectum showed inverse associations among petroleum workers, which were stronger in the mortality studies than in the incidence studies. We found significant elevated risks of childhood leukemia for residents living near petroleum facilities.

The meta-analysis of the nested case-control studies of hematopoietic cancer among petroleum workers in relation to benzene exposure showed some non-significant elevated risk estimates for AML, CLL and leukemia overall but no conclusive exposure–response relationship. Although the present meta-analysis for cohort studies did not suggest elevated risk for kidney cancer, a non-significant increased risk of kidney cancer was observed in the meta-analysis for case-control studies. Among residents living near petroleum facilities there were non-significant elevated results for brain cancer in adults in both cohort (mortality but not incidence) and case control studies, one positive study of urinary bladder cancer, and one positive study of prostate cancer.

The increased risks of mesothelioma, skin melanoma, and multiple myeloma observed among petroleum workers in our review confirm the findings of a previous review [75]. The vast majority of mesothelioma occurs due to asbestos exposure, with even relatively low exposure levels increasing risk [76,77]. Asbestos is widely used for its non-corrosive ability in combustion petroleum pipes in petroleum refineries [78], although there are no available data on the concentration of asbestos fibers in petroleum facilities. Workers who are engaged in maintenance in petroleum companies in particular experienced an increased risk of mesothelioma, probably due to more asbestos exposure [79].

Skin melanoma is also associated with multiple risk factors which workers can be exposed to including occupational chemical exposures (e.g., arsenic) [80] and ultraviolet radiation [81] in outdoor workers, especially the latter as the most likely explanation for the increased risk in petroleum industry workers. Furthermore, the increased risk of multiple myeloma among petroleum employees in the present meta-analysis is supported by additional reviews [10,75] and the epidemiological evidence linking multiple myeloma with hydrocarbon exposure [82,83,84].

However, the increased risk of prostate cancer we observed among petroleum workers was not observed in the review by Wong and Raabe [75], but has already been reported in individual studies, and our systematic review included studies that were published after Wong and Raabe review. The risk of prostate cancer may be accelerated by exposure to cadmium, which is a composite of crude oil classified as carcinogenic to humans [85]. As the increase in risk was observed only for incidence but not mortality, it is unclear how much a major determinant of prostate cancer incidence, namely PSA screening, plays a role, given the medical surveillance workforces normally receive.

The increased risk of incident urinary bladder cancer we observed among petroleum workers was not examined in the review of Schnatter et al. [10]. Instead, Wong and Raabe reported decreased urinary bladder mortality, which was confirmed by our own review. The observed increase in risk could occur as a result of exposure to chemicals such as benzidine which is known to cause bladder cancer [4]. However, this does not explain the reduced risk of urinary bladder cancer mortality. Further studies evaluating disease stage at diagnosis and access to care may help clarify this unexplained discrepancy.

In our review, skin melanoma, prostate cancer and urinary bladder cancer did not show any association in the mortality-based studies. All three cancers have a good prognosis especially in earlier stages and may be detected early through regular medical check-ups; this may explain why we observe an increased incidence of these cancers but not an increased risk of mortality [86,87,88].

The association in childhood leukemia we observed is in agreement with previous reviews [89,90]. Childhood leukemia may occur due to chromosomal translocation of a pre-leukemic clone, in which the genes are rearranged and may be triggered by exposure to environmental factors such as benzene [89]. In adults, a non-significant increased risk of leukemia was observed in the meta-analysis for cohort and case-control studies, but was significantly elevated in stratified analysis of offshore workers when all leukemia subtypes were combined.

The present meta-analysis showed that mortality was reduced for cancers of the lung, esophagus, stomach, colon, rectum, and pancreas, which are known to be associated with life-style factors such as tobacco, alcohol, and diet [91,92]. This suggests that workers in this industry smoke and drink less, eat more healthily and do more physical activity than the general population. Although we observed a significant elevation in stratified analysis among offshore category, as lung cancer is also known to be related to many occupational exposures, lung cancer did not show increased risk or mortality among workers or residents living in proximity to petroleum facilities in the main analysis, even if there was considerable heterogeneity across individual studies. These negative findings are consistent with previous reviews [10,75].

Strengths of this review include that many of the studies had long term follow up in cohort studies, and that both mortality and incidence studies among petroleum workers were clearly separated in the main analyses of our review when compared to previous reviews. Other strengths include stratified analysis by sub-sites of the petroleum industry based on the extracted information. Weaknesses include crude exposure assessment in some studies, e.g., ever employment in this industry, while the exposure variability may be considerable between jobs/tasks and thereby dilute any potential risk for highly exposed workers. Another caveat is the lack of analyses by duration or level of exposure in some of the studies. Insufficient exposure contrasts may have caused bias among petroleum workers, due to non-stratified classes of workers in some of the studies. Furthermore, the cohort studies reported results on multiple cancer sites, for which some analyses were severely under-powered for the detection of any potential risk. Finally, it is relatively common to see lower mortality in occupational cohorts because unhealthy people are usually excluded from employment, which makes the overall death rates lower than those of the general population [93].

The above mentioned limitations could be reduced by improving the exposure assessment, especially in areas with substantial occupational or residential exposures like low and middle income countries (LMICs) with high crude oil production capacity and artisanal refining activities. Unfortunately, there were no studies conducted on petroleum workers and residents living in oil producing communities in Africa. Meanwhile Angola and Nigeria are among the world’s top oil producers. There have been frequent oil spills contaminating landscape and polluting soil and ground water, as well as constant emission of hydrocarbons and other toxic chemicals in the Niger Delta region of Nigeria. In the same region houses are built around petroleum oil wells with pipelines running across the various communities and crude oil is sometimes used for dermal care and ingested for childhood convulsion treatments among other illnesses [94]. The United Nations Environment Programme (UNEP) reported that benzene level was 900 times above WHO guidelines in some spots in Ogoniland, Niger Delta region of Nigeria, as a result of petroleum pollution [95]. Similarly, major oil pollution has also been reported in some other petroleum producing regions of the world like the Middle East and Russian Federation where the oil industry is one key [96,97]. Therefore, well designed epidemiological studies on workers and residents are warranted in these regions.

## 5. Conclusions

In conclusion, living close to petroleum facilities was associated with increased risk of childhood leukemia, while petroleum industry work was associated with an increased risk of mesothelioma, skin melanoma, multiple myeloma, and cancers of the prostate, urinary bladder, and a decreased risk of cancers of the esophagus, stomach, colon, rectum, and pancreas. Offshore petroleum work was associated with an increased risk of lung cancer and leukemia in stratified analysis. Many of the associations, however, appear to be due to factors other than those directly emerging from petroleum production. This applies to asbestos for mesothelioma, which is a rare cancer, so that the number of cases attributable to this risk were small compared to the working population. This also applies to ultraviolet radiation from outdoor work for skin melanoma, or from healthier lifestyles in petroleum workers for cancers of the colon, rectum, esophagus, stomach and pancreas. Prostate cancer incidence studies may have been affected by medical surveillance. For urinary bladder cancer, studies showed inconsistent results, which may not be surprising given that bladder cancer is positively associated with several workplace chemicals (some occurring in the petroleum industry) but also with smoking (apparently reduced in petroleum workers given the reduced risks of other smoking-related cancers), and therefore inclusion criteria of individual studies and their possibilities to adjust for confounders may have had a significant impact on the study results.

The overall evidence remains weak, particularly for the residential studies. Improved exposure assessment is needed in further studies to describe exposure pathways of petroleum and its closest derivatives, e.g., benzene, in order to identify the drivers of the observed cancer risks or identify association which may have been missed using the crude industry-based exposure approach. In particular, there is a need for targeted studies in under-researched areas of high petroleum production with presumably higher exposures. An international consortium guiding new generation studies in Africa, the Middle East and Asia, to harmonize study protocols and exposure assessments, may be the most promising way forward.

## Figures and Tables

**Figure 1 ijerph-18-04343-f001:**
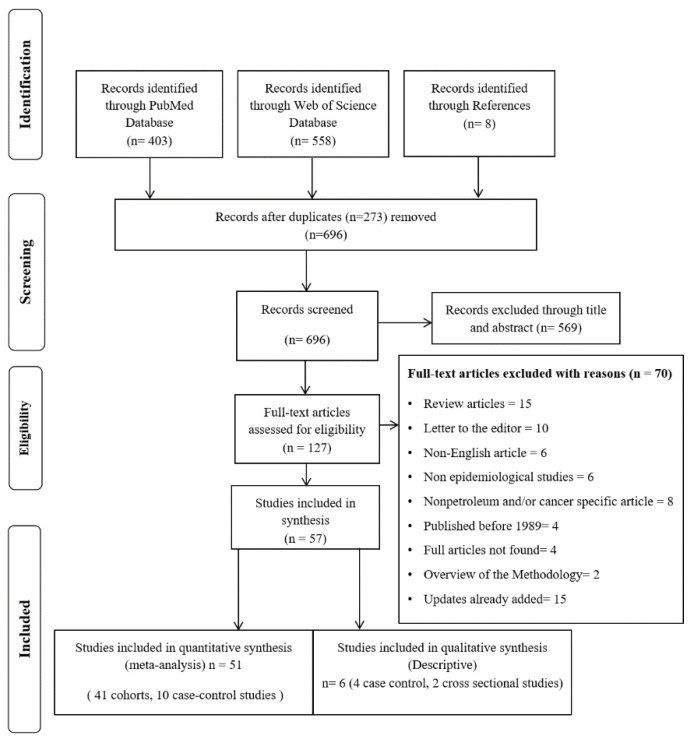
PRISMA Flow Chart of papers included in this systematic review and meta-analysis of cancer incidence and mortality among petroleum industry workers and residents living in oil producing communities.

**Figure 2 ijerph-18-04343-f002:**
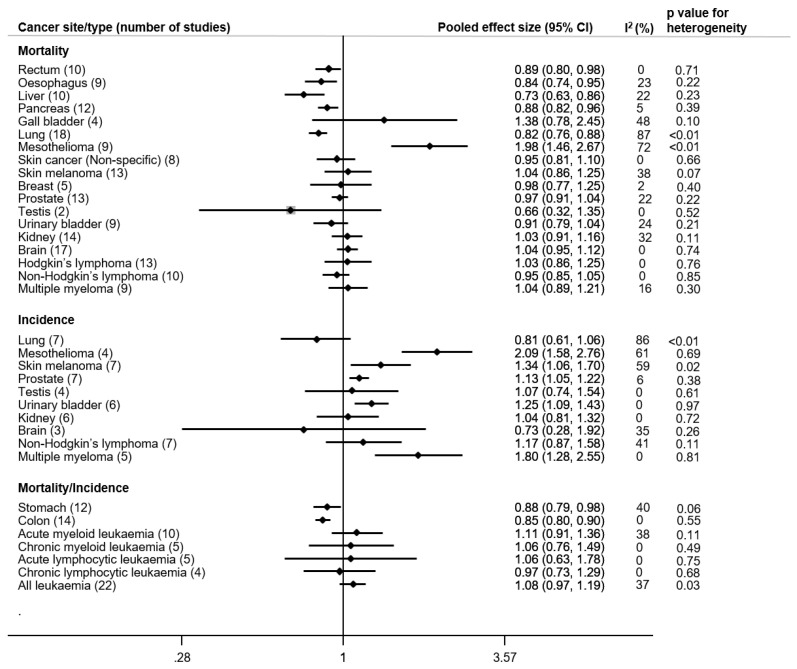
Forest plot of pooled effect sizes and heterogeneity in cohort studies evaluating cancer incidence and mortality among petroleum industry workers sorted by ICD.

**Figure 3 ijerph-18-04343-f003:**
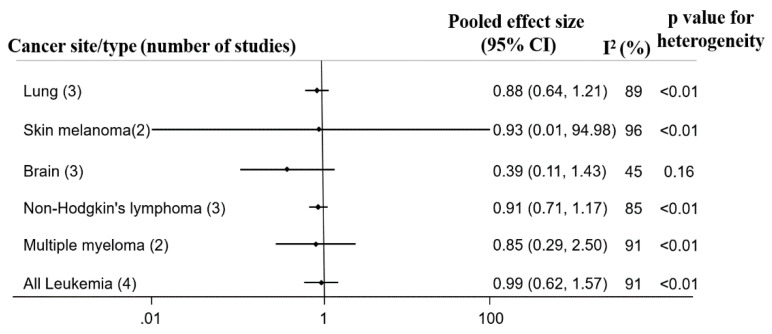
Forest plot of pooled effect size and heterogeneity of cohort studies evaluating cancer relative risk, incidence and mortality among residents of oil producing communities sorted by ICD.

**Figure 4 ijerph-18-04343-f004:**
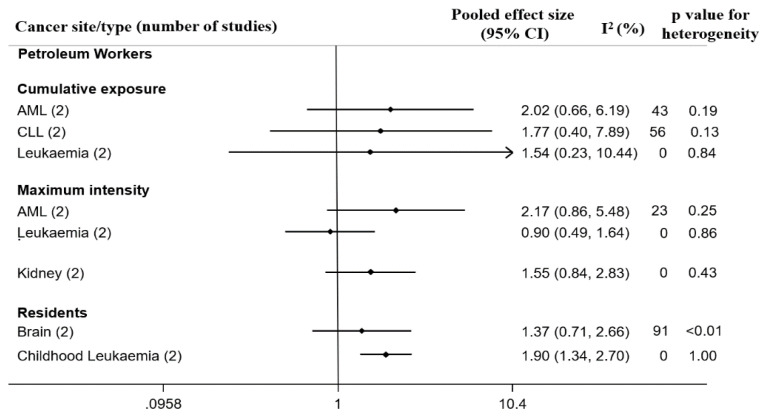
Forest plot of pooled effect sizes and heterogeneity in nested case-control studies evaluating cancer among petroleum industry workers and case-control studies evaluating cancer risk among residents living in proximity to petroleum facilities sorted by ICD.

**Table 1 ijerph-18-04343-t001:** Characteristics of the studies on petroleum workers.

First Author, Year [Reference]	Cancer Site/Type		Country	Minimum Employment Time	Follow-Up (Years)	Exposed Population
**Cohort studies:**						
Christie, 1991 [19]	AML, Brain, Kidney, Leukemia, Lung, Melanoma, MM, NHL, Skin and Urinary bladder		Australia	5 Years	9	>1500
Gun, 2004 [20]	ALL, AML, Brain, CML, Colon, Rectum, Gall bladder, Kidney, Liver, Lung, Mesothelioma, MM, NHL, Esophagus, Pancreas, Prostate, Skin melanoma, Skin, Stomach, Testis and Urinary bladder		Australia	5 Years	16	17,163
Gun, 2006a [9]	Mesothelioma		Australia	5 Years	22	16,543
Gun, 2006b * [21]	ALL, AML, Brain, CML, Colon, Gall bladder, Kidney, Leukemia, Liver, Lung, MM, NHL, Esophagus, Pancreas, Prostate, Skin melanoma, Skin, Stomach, Testis and Urinary bladder		Australia	5 Years	22	17,903
Schnatter, 1993 [22]	Brain		Canada	1 Year	20	6672
Lewis, 2003 [23]	Gall bladder, Kidney, Leukemia, NHL, Prostate and Testis		Canada	1 Month	23	25,292
Schnatter, 2012 [24]	Lung (Incidence studies)		Canada	1 Year	31	17,216
Schnatter, 2019 [25]	AML, Breast, Colon, HL, Kidney, Leukemia, Liver, Lung, Melanoma, Mesothelioma, NHL, Esophagus, Pancreas, Prostate, Rectum, Stomach and Urinary bladder		Canada	1 Year	43	29,379
Gennaro, 1994 [26]	Mesothelioma		Italy	1 Year	77	2300
Consonni, 1999 [27]	Brain, HL, Kidney, Leukemia, Lung, NHL, Skin melanoma, Urinary bladder		Italy	1 Day	43	1583
Pasetto, 2012 [28]	HL, Leukaemia, Lung, Mesothelioma, MM, NHL and Skin melanoma,		Italy	10 Years	43	5627
Bonzini, 2019 [29]	Colon, HL, Liver, Lung, Mesothelioma, MM, NHL, Pancreas, Prostate, Rectum, Skin, Stomach and Urinary bladder		Italy	1 Year	63	5112
Koh, 2011 [30]	Prostate, Skin and Urinary bladder		Korea	1 Day	16	8866
Koh, 2014 [31]	Brain, Colon, Leukaemia, Liver, Lung, NHL, Esophagus, Pancreas, Prostate and Stomach		Korea	NR	6	14,698
Aas, 2009 [32]	AML, Leukaemia, Lung, Mesothelioma, Prostate and Skin melanoma		Norway	20 Days	41	28,000
Stenehjem, 2014 [33]	Lung, Mesothelioma, Skin melanoma and Urinary bladder		Norway	20 Days	11	41,140
Kirkeleit, 2008 [34]	ALL, AML, CLL, CML and MM		Norway	<1 Year	22	27,919
Jarvholm, 1997 [35]	Brain, Kidney, Leukaemia, Lung, Prostate, and Skin		Sweden	1 Year	34	4319
Rushton,1993 [36]	Brain, HL, Kidney, Lung, Leukaemia and MM		UK	1 Year	40	23,306
Sorahan, 2007 [37]	AML, Breast, Colon, Gall bladder, HL, Kidney, Leukaemia, Liver, Lung, Mesothelioma, MM, Esophagus, Pancreas, Prostate, Rectum, Skin, Stomach, Testis and Urinary bladder		UK	1 Year	53	45,032
Satin, 1996 [38]	AML, Brain, Breast, HL, Kidney, Leukaemia, Lung and Skin melanoma,		USA	1 Day	51	17,844
Divine,1999a [39]	Brain, Lung and Mesothelioma		USA	1 Year	47	28,840
Gamble, 2000 [40]	Kidney, Leukaemia, Lung, Mesothelioma, Prostate and Skin melanoma		USA	1 Month	23	6238
Huebner, 2000 [41]	Leukaemia, MM and NHL		USA	1 Day	12	8942
Sathiakumar, 2001 [42]	Brain, Colon, Kidney, Lung, NHL, Prostate, Skin melanoma, Testis and Urinary bladder		USA	NR	12	5641
Wong, 2001a [43]	Brain, Breast, Colon, HL kidney, Leukaemia, Lung, Esophagus, Prostate, Skin melanoma		USA	1 Year	39	3328
Satin, 2002 [44]	ALL, AML, CLL, CML, Colon, HL, Kidney, Leukaemia, Liver, Lung, MM, NHL, Esophagus, Pancreas, Rectum, Skin melanoma and Stomach		USA	1 Year	46	18,512
Tsai 2003 [45]	Kidney, Leukaemia, Liver, Lung, Prostate and Rectum		USA	6 Months	31	4221
Huebner, 2004 [46]	Brain and Skin melanoma		USA	1 Month	28	14,644
Buffler, 2004 [47]	Brain		USA	6 Months	32	3779
Tsai, 2004 [48]	Brain, Colon, Kidney, Leukaemia, Lung, MM, NHL, Pancreas, Prostate, Skin melanoma, Stomach and Urinary bladder		USA	>6 Months	12	4639
Tsai, 2007 [49]	Brain, Colon, HL, Kidney, Leukaemia, Liver, Lung, NHL, Esophagus, Pancreas, Prostate, Rectum, Skin melanoma and Stomach		USA	3 Months	56	10,621
Huebner2009 [50]	ALL, AML, Breast, CLL, CML, Colon, HL, Kidney, Leukaemia, Liver, Lung, Mesothelioma, MM, NHL, Esophagus, Pancreas, Prostate, Rectum, Skin melanoma, Skin, Stomach and Urinary bladder		USA	1 Day	22	127,266
Divine, 1999b [51]	HL, Kidney, Leukaemia, Pancreas, Prostate, Skin, Stomach and Urinary bladder		USA	1 Year	47	28,480
Wong, 2001b [52]	Brain, Colon, HL, Kidney, Leukaemia, Lung, Pancreas, Rectum, Skin melanoma and Stomach		USA	1 Year	43	7543
**Nested Case-control studies**		**Exposure**			**Cases (*n*)**	**Controls (*n*)**
Finkelstein, 1996 [53]	Mesothelioma	Asbestos	Canada	NR	17	46
Schnatter,1996 [54]	Leukaemia	Benzene	Canada	NR	14	55
Anttila, 2015 [55]	Kidney	Hydrocarbons	Finland	3 Months	30	81
Rushton, 2014 [56]	AML, CLL	Benzene	Intercontinental	I Year	140	568
Glass, 2014 [57]	CML	Benzene	Intercontinental	NR	28	122
Stenehjem, 2015 [58]	AML, CLL and MM	Benzene	Norway	20 Days	91	415
Rushton,1997 [59]	Leukaemia	Benzene	UK	I Year	91	364
Poole, 1993 [60]	Kidney	Hydrocarbons	USA	6 Months	100	406

Abbreviations: HL—Hodgkin’s lymphoma, NHL—Non-Hodgkin’s lymphoma, AML—Acute myeloid leukemia, MM—Multiple myeloma, CML—Chronic myeloid leukemia, CLL—Chronic lymphocytic leukemia. * About 3000 new refinery workers were part of the update.

**Table 2 ijerph-18-04343-t002:** Characteristics of studies of residents living near petroleum facilities.

Author and Year	Cancer Site/Type	Type of Study	Country	Data Source	Follow-Up/Years of Exposure	Exposed Population
**Cohort studies**						
Hurtig, 2002 [61]	Brain, HL, Lung and Skin melanoma	Cohort	Ecuador	National records	4	~280,000
Hurtig, 2004 [62]	Leukaemia	Cohort	Ecuador	National records	20	~356,406
Kelsh, 2009 [63]	Brain, Lung, Leukaemia, MM and NHL and Skin melanoma	Cohort	Ecuador	National records	6	15335
Ramis, 2012 [64]	NHL	Cohort	Spain	National records	10	1,744,988
Barregard, 2009 [65]	Leukaemia	Cohort	Sweden	National records	30	15,000
Sans, 1995 [66]	HL, Leukaemia, Lung, MM and NHL	Cohort	UK	Research Group Population Censuses and Welsh cancer registry	18	115,721
**Cross sectional studies**						
Choi, 2018 [67]	Prostate	Cross sectional	Korea	Divisional and National records		63,042
Lyons, 1995 [68]	Leukaemia	Cross sectional	UK	Wales and National records		2632
**Case control studies**					**Cases (*n*)**	**Controls (*n*)**
Tsai, 2009[69]	Urinary bladder	Case control	China/Taiwan	Bureau of Vital Statistics of the Taiwan	821	821
Weng, 2008 [70]	Leukaemia	Case control	China/Taiwan	Bureau of Vital Statistics of the Taiwan	405	405
Liu, 2008 [71]	Brain	Case control	China/Taiwan	Bureau of Vital Statistics of the Taiwan	340	340
Yu, 2005 [72]	Brian	Case control	China/Taiwan	Taiwanese population registry data	143	364
Micheli, 2014 [73]	Hematological malignancies	Case control	Italy	Italian National Institute of Statistics	177	349
McKenzie, 2017 [74]	Leukaemia	Case control	USA	Colorado cancer registry	138	528

Abbreviations: HL—Hodgkin’s lymphoma, NHL—Non-Hodgkin’s lymphoma MM—Multiple myeloma.

**Table 3 ijerph-18-04343-t003:** Stratified/subgroup of cancer type by industry category.

Cancer Site/Type	*n*	Refinery OR 95% CI	*n*	Petroleum OR 95% CI	*n*	Petrochemical OR 95% CI	*n*	Refinery and Petrochemical OR 95% CI	*n*	Offshore OR 95% CI
Respiratory system										
Lung	8	0.83, 0.76–0.91	7	0.73, 0.63–0.85	2	0.57, 0.32–1.02	7	0.80, 0.67–0.96	3	1.20, 1.03–1.39
Mesothelioma	4	1.94, 1.00–3.76	5	1.58, 1.30–1.93		x	2	2.86, 2.16–3.77	2	2.47, 1.66–3.67
Skin										
Skin melanoma	7	1.15, 0.88–1.50	6	1.28, 1.10–1.50	2	0.85, 0.24–3.06	7	0.84, 0.55–1.27	2	1.44, 0.68–3.02
Skin cancer (Non-specific)	2	1.18, 0.71–1.95	5	0.89, 0.75–1.06		x	2	1.12, 0.72–1.74		x
Urinary tract										
Urinary bladder	3	1.00, 0.88–1.14	7	0.93, 0.69–1.25		x	4	1.08, 0.78–1.49		x
Kidney	8	1.10, 0.98–1.24	7	0.91, 0.77–1.06		x	6	1.00, 0.79–1.25		x
Reproductive system										
Prostate	4	1.01, 0.89–1.15	8	0.97, 0.83–1.146		x	7	1.01, 0.93–1.11		x
Testis	2	0.63, 0.28–1.42	4	1.00, 0.68–1.48		x		x		-
Breast	5	1.00, 0.78–1.27		x		x		x		-
Digestive and accessory digestive organ										
Esophagus	5	0.91, 0.81–1.03	3	0.67, 0.56–0.80		x	2	0.88, 0.59–1.30		-
Stomach	5	0.96, 0.83–1.10	4	0.79, 0.68–0.92		x	4	0.76, 0.60–0.95		-
Colo-rectal	12	0.87, 0.80–0.94	7	0.81, 0.74–0.88		x	6	0.87, 0.76–0.98		-
Liver	4	0.63, 0.36–1.08	4	0.73, 0.62–0.86		-	3	0.76, 0.56–1.02		-
Gall bladder	2	1.08, 0.45–2.56	3	1.83, 0.69–4.82		-		-		-
Pancreas	5	0.88, 0.72–1.07	4	0.84, 0.75–0.95		-		0.94, 0.78–1.13		-
Central Nervous System										
Brain	7	1.00, 0.88–1.13	7	1.02, 0.89–1.16		x	6	1.05, 0.78–1.42		-
Hematological										
Hodgkin’s lymphoma	8	1.07, 0.85–1.34	2	1.11, 0.75–1.63		x	3	0.86, 0.43–1.71		-
Non-Hodgkin’s lymphoma	2	0.96, 0.74–1.23	8	0.97, 0.87–1.09	2	1.09, 0.35–3.32	5	1.04, 0.63–1.72		-
Multiple Myeloma	5	1.03, 0.82–1.30	7	1.26, 0.94–1.68		x		x		x
Leukaemia	16	1.07, 0.98–1.17		1.07, 0.91–1.26		x	8	1.02, 0.80–1.29	6	1.47, 1.12–1.92

- = No study, x = Only one study.

## Data Availability

Data is contained within the article and Appendix A.

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
