# Peer review of "Cancer Incidence and Mortality among Petroleum Industry Workers and Residents Living in Oil Producing Communities: A Systematic Review and Meta-Analysis"

_ijerph, 2021, doi:10.3390/ijerph18084343_

Round 1

Reviewer 1 Report

Thank you for giving me the opportunity to review this paper. It is an elegant systematic review and meta-analysis aimed at evaluating the cancer risk in petroleum-exposed workers and in residents living near petroleum facilities

The meta-analysis is performed in accordance with good standards. I do not have any comments regarding the methodology of the work performed. The only suggestion I have is that several types of “petroleum facilities” may be distinguished, and for sake of having more homogenous results I would suggest group studies by type of them.

The minor aspects:

The paper is rather long and difficult to follow. I would recommend t skip the part related to the study of the general population leaving closed to petroleum facilities.

On the other hand, I would recommend adding in the introduction a paragraph in which main exposures occurring in petroleum facilities take place.  Potential exposure to asbestos, as the cause of mesothelioma, needs to more strongly addressed.

In conclusions it is stated that among petroleum workers, five cancer types were associated with elevated risks, namely mesothelioma in both incidence and mortality studies, as well as in incidence (but not mortality) studies multiple myeloma, skin melanoma, prostate cancer, and urinary bladder cancer. It should be made very much clear that mesothelioma is an extremely rare cancer, and the number of cases that could be attributable to this risk would be very small.

It should be made also clear that based on the literature, even very low exposure to asbestos may increase the risk of that deadly cancer. On other hand,  have anyone measured the asbestos fibers concentration in “petroleum facilities”?  Authors provide information that, asbestos was widely used for its non-corrosive ability in combustion petroleum pipes in petroleum refineries, however, no measurements exposure date was provided.

Regarding the cancers of the liver, lung, oesophagus, colon, pancreas, stomach, and rectum which showed inverse associations among petroleum workers,  I would suggest, the authors explained what is the implications of these results. 

Author Response

Response to Reviewer 1 Comments

The authors appreciate the time the reviewers took to review and provide useful feedback to improve the quality of manuscript. We addressed each comment carefully and provided responses below.

Point 1: Thank you for giving me the opportunity to review this paper. It is an elegant systematic review and meta-analysis aimed at evaluating the cancer risk in petroleum-exposed workers and in residents living near petroleum facilities

The meta-analysis is performed in accordance with good standards. I do not have any comments regarding the methodology of the work performed. The only suggestion I have is that several types of “petroleum facilities” may be distinguished, and for sake of having more homogenous results I would suggest group studies by type of them.

Response 1: We like to thank the reviewer for the supportive assessment and the constructive comments. The reviewer’s suggestions have been addressed. We conducted stratified/subgroup analysis as suggested (See new Table 3). 

Point 2: The minor aspects:

The paper is rather long and difficult to follow. I would recommend to skip the part related to the study of the general population leaving closed to petroleum facilities.

On the other hand, I would recommend adding in the introduction a paragraph in which main exposures occurring in petroleum facilities take place.  Potential exposure to asbestos, as the cause of mesothelioma, needs to more strongly addressed.

Response 2: Authors appreciate the suggestion of the reviewer. However, the residential aspect of the research is an integral component as we hope that it will rekindle community awareness and advocacy for safer environment. We request that it should not be dropped or skipped.

As suggested by the reviewer, the introduction section has been improved to show main exposures and potential exposure to asbestos, as the cause of mesothelioma (line 37-43).

Point 3: In conclusions it is stated that among petroleum workers, five cancer types were associated with elevated risks, namely mesothelioma in both incidence and mortality studies, as well as in incidence (but not mortality) studies multiple myeloma, skin melanoma, prostate cancer, and urinary bladder cancer. It should be made very much clear that mesothelioma is an extremely rare cancer, and the number of cases that could be attributable to this risk would be very small.

Response 3: The authors agree with the reviewer’s suggestion. The suggested statements by the reviewer have been incorporated in the manuscript text (conclusion section, line 623-626).

Point 4: It should be made also clear that based on the literature, even very low exposure to asbestos may increase the risk of that deadly cancer. On other hand, have anyone measured the asbestos fibers concentration in “petroleum facilities”?  Authors provide information that, asbestos was widely used for its non-corrosive ability in combustion petroleum pipes in petroleum refineries, however, no measurements exposure date was provided.

Response 4: The suggested statements by the reviewer have also been incorporated in the manuscript text (discussion section, line 534-535).  We were unable to provide information on measurements since there was no available data on the measurement of asbestos fibers concentration in petroleum facilities in the studies reviewed.

Point 5: Regarding the cancers of the liver, lung, oesophagus, colon, pancreas, stomach, and rectum which showed inverse associations among petroleum workers, I would suggest, the authors explained what is the implications of these results. 

Response 5: The authors agree with the reviewer that it is also important to discuss about the implications of inverse associations. We stated in the discussion section that those cancers are known to be associated with life-style factors such as tobacco, alcohol, and diet, which may indicate that workers in this industry smoke and drink less, eat healthier and have more physical activity than the general population.

Reviewer 2 Report

The manuscript titled “Cancer incidence and mortality among petroleum industry workers and residents living in oil producing communities: A systematic review and meta-analysis” presents an interesting review on epidemiological evidence about the possible detrimental effect of both occupational and residential exposure to petroleum-related facilities. In general the manuscript is well-written, methods are almost clear and complete. Some major revisions are recommended especially about sensitivity and/or stratified analysis in order to improve the quality and insight of data presentation and analysis before publication Major comments:

- L59-61. It is not entirely clear the reason to restrict period of study publication from 1990 in the eligibility criteria. There are no older studies or they were excluded? In the latter case, why?

- In addition, about the inclusion criteria, it is not clear the definition of petroleum facilities, since also oil and gas development could be included, but some studies are missing (e.g. PMIDs: 27783932 about residential exposure, or PMID: 17906934 on workers). Please clarify this issue through better specification of eligible criteria and/or check for missed studies.

- L129-131: please consider to avoid the use of fixed cutpoint for interpretation and reporting of results according the most recent literature on the topic (see for example the 2016 ASA Statement - DOI: 10.1080/00031305.2016.1154108, and other recent papers - PMIDs: 2898712, 27209009, 27272951, 28938715, 29650628). In particular, about the interpretation of Egger test, please consider some recent recommendation including the assessment of its magnitude along with its precision (PMID: 31273125).

- In Figure 2, instead of increasing ES, it may be helpful to present summary ES by cancer site or system (as they are discussed in the text). In addition as regards study design, did authors assess possible differential results from cohort studies compared to case-control or cross-sectional. This would strengths review findings, especially in case of contrasting results. Finally, did author perform analysis restricted to high quality studies. Assessment of risk of bias has been performed, but it does not seem it has been used in the analysis. Please clarify.

Minor comments:

- I-squared statistics may be not displayed in the Abstract.

- L69: it could be clarified the reason e.g. with the longest follow-up or the bigger population.

- The database searches could be included as supplemental material taking them out of the main text.

Author Response

Response to Reviewer 2 Comments

The authors appreciate the time the reviewers took to review and provide useful feedback to improve the quality of manuscript. We addressed each comment carefully and provided responses below.

Point 1: The manuscript titled “Cancer incidence and mortality among petroleum industry workers and residents living in oil producing communities: A systematic review and meta-analysis” presents an interesting review on epidemiological evidence about the possible detrimental effect of both occupational and residential exposure to petroleum-related facilities. In general the manuscript is well-written, methods are almost clear and complete. Some major revisions are recommended especially about sensitivity and/or stratified analysis in order to improve the quality and insight of data presentation and analysis before publication

Response 1: We thank the reviewer for the supportive assessment and for the constructive comments. As suggested, we conducted stratified/subgroup analysis (Please see new Table 3).

Point 2: Major comments:- L59-61. It is not entirely clear the reason to restrict period of study publication from 1990 in the eligibility criteria. There are no older studies or they were excluded? In the latter case, why?

Response 2: Indeed, the vast majority of publications is from 1990 onwards. Some of the studies published earlier were updated after 1990 and were included in the present review. These studies have therefore a long follow up duration of 31 to 77 years. Other than those studies, very few studies were published before 1990, on different working circumstances. We felt that including those studies would rather dilute the picture than adding scientific insight. Therefore, we focused on the publications from 1990. We added this clarification in Line 65-69

Point 3: In addition, about the inclusion criteria, it is not clear the definition of petroleum facilities, since also oil and gas development could be included, but some studies are missing (e.g. PMIDs: 27783932 about residential exposure, or PMID: 17906934 on workers). Please clarify this issue through better specification of eligible criteria and/or check for missed studies.

Response 3: To improve the clarify of our eligibility criteria, we added a definition of petroleum facilities in the method section (Line 87-89). Based on the eligibility criteria described in Line 69-74, PMIDs: 27783932 about residential exposure, was not eligible. PMIDs: 27783932 fall short of this, as it is a review of non-quantitative data. We only included cohort, case control and cross-sectional articles with odd ratios and 95% CI. However, PMID: 17906934 on workers met the criteria and have been included and analyzed accordingly. Author deeply appreciate the reviewer for this contribution

Point 4: - L129-131: please consider to avoid the use of fixed cutpoint for interpretation and reporting of results according the most recent literature on the topic (see for example the 2016 ASA Statement - DOI: 10.1080/00031305.2016.1154108, and other recent papers - PMIDs: 2898712, 27209009, 27272951, 28938715, 29650628). In particular, about the interpretation of Egger test, please consider some recent recommendation including the assessment of its magnitude along with its precision (PMID: 31273125).

Response 4: Authors did not use fixed cutpoint for the interpretation or for magnitudes of heterogeneity. The PMID: 31273125 recommended stated the use of “four levels, i.e., unimportant, moderate, substantial, and considerable; each level contained roughly 30% of the distribution”.

However, we also used the same four levels:

  1. “probably unimportant” when I2 was 1% to 35%,
  2. “moderate” when I2 was 36% to 55%,
  3. “substantial” when I2 was 56% to 70% and
  4. “considerable” when I2 was 71% to 100%.

(Line 131-134)

Point 5: - In Figure 2, instead of increasing ES, it may be helpful to present summary ES by cancer site or system (as they are discussed in the text).

In addition as regards study design, did authors assess possible differential results from cohort studies compared to case-control or cross-sectional. This would strengths review findings, especially in case of contrasting results.

Finally, did author perform analysis restricted to high quality studies. Assessment of risk of bias has been performed, but it does not seem it has been used in the analysis. Please clarify.

Response 5: The authors agree with the reviewer’s suggestion, and have ordered the ES according to ICD

Yes, authors assessed the cancer that appeared on both cohort and case control studies and were discussed accordingly. I.e. Kidney (line 491-493), brain (line 494-495) and leukaemia (line 536-537)

The only set of studies with low scores were cohort studies of residents (average 41%). We did not exclude this set of papers based on the methodological quality due to very few papers being eligible for the final analysis. This have been stated in the manuscript (Line 172-174) and it was also emphasized in the conclusion section of the manuscript (Line 596)

Point 6: Minor comments:

- I-squared statistics may be not displayed in the Abstract.

- L69: it could be clarified the reason e.g. with the longest follow-up or the bigger population.

- The database searches could be included as supplemental material taking them out of the main text.

Response 6:

As suggested, all I-squared statistics have been deleted from the Abstract.

The suggested clarification has been made in the manuscript (Line 77-78)

As suggested by the reviewer, the search strategy has been taken to supplementary material and referred to accordingly.

Round 2

Reviewer 2 Report

Authors adequately addressed all issues raised during the first review and the manuscript has much improved with all my previous concerns solved or entirely clarified in the response and in the main text.